# NANOBODY^®^ Molecule, a Giga Medical Tool in Nanodimensions

**DOI:** 10.3390/ijms241713229

**Published:** 2023-08-25

**Authors:** Sarah Kunz, Manon Durandy, Laetitia Seguin, Chloe C. Feral

**Affiliations:** 1Université Côte d’Azur, CNRS UMR7284, INSERM U1081, IRCAN, 06107 Nice, France; sarah.kunz@univ-cotedazur.fr (S.K.); manon.durandy@univ-cotedazur.fr (M.D.); laetitia.seguin@inserm.fr (L.S.); 2Department of Oncology, Sanofi Research Center, 94400 Vitry-sur-Seine, France

**Keywords:** single variable domain, VHHS, diagnostic and therapeutic strategies

## Abstract

Although antibodies remain the most widely used tool for biomedical research, antibody technology is not flawless. Innovative alternatives, such as Nanobody^®^ molecules, were developed to address the shortcomings of conventional antibodies. Nanobody^®^ molecules are antigen-binding variable-domain fragments derived from the heavy-chain-only antibodies of camelids (VHH) and combine the advantageous properties of small molecules and monoclonal antibodies. Nanobody^®^ molecules present a small size (~15 kDa, 4 nm long and 2.5 nm wide), high solubility, stability, specificity, and affinity, ease of cloning, and thermal and chemical resistance. Recombinant production in microorganisms is cost-effective, and VHH are also building blocks for multidomain constructs. These unique features led to numerous applications in fundamental research, diagnostics, and therapy. Nanobody^®^ molecules are employed as biomarker probes and, when fused to radioisotopes or fluorophores, represent ideal non-invasive in vivo imaging agents. They can be used as neutralizing agents, receptor-ligand antagonists, or in targeted vehicle-based drug therapy. As early as 2018, the first Nanobody^®^, Cablivi (caplacizumab), a single-domain antibody (sdAb) drug developed by French pharmaceutical giant Sanofi for the treatment of adult patients with acquired thrombocytopenic purpura (aTTP), was launched. Nanobody^®^ compounds are ideal tools for further development in clinics for diagnostic and therapeutic purposes.

## 1. Introduction

Since their discovery more than three decades ago, antibodies (Abs) have transformed the clinical field and have become the predominant class of new drugs developed for therapeutic purposes in recent years. To date, more than 100 monoclonal Abs have been marketed or are currently being tested for the treatment of various human diseases, including many cancers, autoimmune diseases, and infectious diseases. Thus, Abs are very successful drugs, and the biopharmaceutical industry has a long track record and knowledge base for the development of Abs. Nonetheless, some functional drawbacks remain, such as suboptimal pharmacokinetics, insufficient tissue accessibility, and impaired interaction with the immune system. In addition, the manufacturing and purification processes cause limitations in production capacity and a cost increase [1]. Thus, alternative and, potentially, complementary options to therapeutic Abs would greatly improve treatment opportunities [2].

The discovery of heavy-chain antibodies, a camelid-specific form of Abs, has raised high hopes. Abs are known to be composed of two identical H-chains (heavy) and L-chains (light) in all mammalian species. Nonetheless, sera from camelids, including dromedary, camel, and llama, contain unusual Abs that do not incorporate light chains (Figure 1). This 1989 discovery from Cécile Casterman and her husband, Raymond Hamers, teachers at the Vrije Universiteit Brussel (VUB), occurred while training students on how to obtain Abs from the blood of dromedaries to develop a serodiagnostic test for trypanosome infection in camelids. Camelid sera express unique types of Abs composed only of two identical heavy chains and called “heavy-chain-only antibodies” (HCAbs). The variable domain of HCAb, corresponding to the paratope, also known as an antigen-binding site, is called camelid heavy-chain variable domains (VHH) and, more recently, Nanobody^®^ molecule, a registered trademark of Ablynx NV [3,4,5,6,7,8]. Over the years, beneficial properties of VHHs have emerged, including their small size, good tissue penetration, high specificity, affinity, solubility and stability, ease of development, low immunogenicity, and inexpensive production (Figure 1). Thus, these attractive molecules pave the way to novel and complementary therapeutic and diagnostic approaches with great potential for human health.

## 2. Structure, Generation, Comparison and Advantages

### 2.1. Structure of Nanobody^®^ Molecules

Conventional Abs are big molecules (150 kDa) composed of two identical heavy chains and two identical light chains connected by disulfide bonds and shaped like a capital letter “Y” (Figure 2A). Each heavy and light chain is formed by different domains: (i) constant domains (CL, CH1, CH2, CH3) promoting the interaction with immune cells, such as macrophages or mast cells; (ii) variable domains (VH for heavy chain and VL for light chain) at the N-terminal of each chain, capable of recognizing antigens and called Fab (fragment antigen binding). This region is called “variable” because it differs in the amino acid sequence composition for each antigen. Each variable domain is composed of three hypervariable regions called the complementarity determining regions (CDRs). The diversity of binding specificity is created by the VH-VL combination. The specificity (recognition) and affinity (binding) of the Abs to their target are dependent on six specific CDR loops, differing in length and sequence between Abs (Figure 2A).

Camelids produce these conventional Abs, composed of a heterodimeric tetramer comprised of heavy and light chains, but they also produce Abs that do not incorporate light chains, which contribute to around 50% of all camelid serum IgGs. These heavy-chain-only antibodies (HCAbs) are present in a different fraction than conventional Abs (Figure 2B). They can be easily isolated from their serum using affinity chromatography combined with differential adsorption on protein A and protein G [3]. This gives access to three fractions containing IgGs of distinct molecular weight: the so-called “IgG-1” subclass contains the conventional antibodies comprising two heavy and two light chains; the “IgG-2” and “IgG-3” subclasses contain HCAbs composed of heavy chains. All IgGs from camels bind to protein A. While IgG-2 subclasses do not adsorb on protein-G and can be collected in flow-through, IgG-1 and IgG-3 can be eluted with a specific buffer. The most notable differences between conventional Abs and HCAbs are (Figure 2B): (i) the absence of the first constant domain (CH1) in the heavy chain, which in a conventional Ab connects to the light chain and interacts with the VH domain. Sequence analysis revealed that the splice site, immediately downstream of the CH1 exon, is disrupted by point mutations at highly conserved amino acids, resulting in the removal of the entire CH1 domain [9]; and (ii) the absence of the light chain.

Interestingly, the variable domains of the HCAbs are adapted to bind antigens in the absence of the VL [10]. Indeed, the Fab fragment involved in antigen-binding is replaced by a specific domain called the VHH, which is as functional as the Fab in conventional Abs (Figure 2B). The VHH is composed of four conserved regions called the frameworks (FRs) connected by three hypervariable regions referred to as CDRs (CDR1, CDR2, and CDR3) (see above). Among the CDRs, CDR3 is the main contributor to binding specificity, while CDR1 and CDR2 assist in binding strength [11,12]. Compared to conventional Ab, VHH can have a longer CDR3, which provides them with more diverse paratopes. In addition, the structural organization of the FR and VH regions is similar to the VH in conventional Abs, but with few differences in the FR2 and CDRs. Some highly conserved amino acids are naturally substituted in the VHH. These amino acids are hydrophobic in conventional Abs and allow interaction with the VL domain. Conversely, in VHH, the amino acids are hydrophilic and block association with the VL domain. The nature of these substitutions is consistent with the increased solubility behavior of an isolated camel VHH domain [10,13,14].

### 2.2. Generation of Nanobody^®^ Molecules

The development of therapeutic Abs requires low-cost and high-yield production. Therefore, improvement of their stability, solubility, size, and ease of handling is still in progress for biotechnology diagnosis and human therapy applications. Nowadays, a variety of molecules besides conventional immunoglobulin molecules are being developed to optimize their size and stability, such as the antigen-binding region fragment (Fab), responsible for antigen recognition and binding specificity, or the development of single-chain variable fragments (scFvs), consisting of the light and heavy chain variable domains. To date, VHHs represent a great candidate for this low-cost and high-yield strategy.

Three types of libraries have been developed to generate antigen-specific VHHs: immune, naive, and synthetic libraries (Figure 3). For the immune library, VHHs are easily produced by immunization of camelids with the target antigen. Transgenic mouse models engineered to produce HCAbs by their B cells were generated and could serve as an alternative host for immunization [15,16]. The naive library is generated by amplification of the variable region of HCAb from peripheral blood mononuclear lymphocytes of camels that have not been immunized yet. While this method allows the identification of VHHs binding to non-immunogenic antigens, it requires a mix of multiple animals to avoid bias due to a previous infection [17,18]. In the synthetic library, a specific VHH scaffold is used as the basic scaffold. They are generated by either using a customized trinucleotide approach leading to diversity in the 3 CDRs or by using degenerated codon NNK to randomize the library by introducing CDR3 diversity. Then, the DNA of the individual VHH regions is assembled by PCR to form a complete VHH gene. While immune libraries produce VHHs with great specificity and affinity for their targets, naive and synthetic libraries can generate large and diverse VHHs (Figure 3).

To select specific VHHs from a library, many methods have been developed, such as bacterial display [19], yeast display [20,21], intracellular 2-hybrid selection, ribosome display [22], or phage display, which remains the most robust technique. Phage display is an in vitro technology developed in 1985 and used to identify specific polypeptide chains displayed on the surface of bacteriophages according to their affinity for absorption [23]. Phage display workflow is a multistep process consisting of mRNA extraction from camelid immune cells, cDNA synthesis by reverse transcription, VHH amplification by polymerase chain reaction (PCR), and ligation into a phagemid (phage M13 is the most commonly used), which is subsequently transformed into host cells such as *E. coli* to generate a VHH library. Then, the library is infected with a helper phage to generate phage particles containing the VHH coding sequence. To select antigen-specific phage-displaying VHHs, the display library is incubated with an immobilized target so that antigen-specific VHHs at the surface of the phage remain bound during extensive washing to remove non-reactive phages. Phages that exhibit the highest affinity are selected for another round of panning, and high-affinity VHHs are usually identified after 3 to 5 rounds of biopanning. Then, they are eluted by pH shock or high salt to be further characterized (Figure 3). Phage display allows for large production of VHHs with great diversity.

### 2.3. Comparison of Nanobody^®^ Molecules with Other Types of Therapeutic Antibodies

Over the years and with the success of mAb therapy, attempts have been made to break mAb into fragments to improve their tissue penetration and facilitate their production. Small Ab fragments such as Fab (antigen-binding fragments) and scFv (single chain fragment variable) have been developed as reliable options for clinical applications. Fab agents are the oldest class of mAb fragment therapeutics. Several Fab products have already been approved by the US Food and Drug Administration (FDA) for clinical use (certolizumab pegol, ranibizumab, abciximab, and idarucizumab) [24,25]. They contain one constant and one variable heavy and light chain domain, respectively. Their small size (50 kDa) allows them to penetrate tissues more efficiently and be cleared from the blood more rapidly than mAb. Since they lack Fc fragments, they do not interfere with anti-Fc-mediated Ab detection. However, Fabs are less stable than mAbs and aggregate easily. In addition, production of Fabs by chemical and protease digestion of full-size Ab is difficult, expensive, and requires a large amount of Ab starting material [26]. In the past decade, scFv has become a very attractive fragment used as a key element of most bispecific antibodies (BsAb) [27]. The scFv fragment contains only the variable domains of the heavy and light chains of a mAb connected by a flexible linker. Tandem scFv can be used to target two different antigens on two different cells, two different antigens on the same cell, or two different epitopes on the same antigen. The most widely used BsAb is a bispecific T-cell engager (BiTE), which enhances the patient’s immune response to tumors by retargeting T-cells to tumor cells. BiTEs consist of two scFvs, one of which binds to a T-cell-specific molecule, usually CD3, and one of which binds to a tumor cell antigen, resulting in T-cell activation and tumor cell death [27]. In 2014, blinatumomab, a BiTE that targets CD19 and CD3, was the first scFv-based therapy approved by the FDA for relapsed and refractory B-cell precursor acute lymphoblastic leukemia. Several BiTEs targeting different tumor-associated antigens have been developed and are currently in clinical trials [28]. scFv and VHH share similar advantages such as great affinity, small size, lack of Fc region leading to low immune response, ease of engineering, and inexpensive cost to generate by phage display and production in bacteria [29,30]. However, many drawbacks arising from the structure of scFv, including lower solubility and a tendency to aggregate due to four residues (V37, G44, L45, and W47) in the FR2 region that create a hydrophobic region, a preference for linear epitopes due to their shorter CDR3, as well as lower thermostability and robustness, highlight the tremendous benefits of nanobody molecules as therapeutic alternatives.

Altogether, research, clinical, and industry enthusiasm for VHHs is driven by their unique properties. Without being mAb substitutes, VHHs can clearly be an outstanding alternative, bringing different advantages (detailed in Table 1). VHHs present exciting features that are highly complementary to mAbs and could eventually lead to synergistic therapeutic benefits (Table 1).

### 2.4. Advantages of Nanobody^®^ Molecules

A key feature of VHHs is their small size (15 kDa), which allows them to penetrate tissue and reach target cells quickly and deeply, while Abs’ large size (150 KDa) prevents efficient diffusion through tissue and remains a major hurdle limiting their efficacy [38]. Thus, VHH technology opens new perspectives in diagnostics and therapy.

Extracellular matrix (ECM) deposition is characteristic of many diseases, including fibrosis, cardiovascular disorders, and cancer [46]. Efforts to modulate the ECM stiffness could improve treatment response, and several studies have shown that ECM softening improves response to chemotherapy and immunotherapy [47,48,49]. However, efficacy remains limited. In this context, VHH could represent a fantastic opportunity. As an example, VHH targeting Tenascin-C, a component of the ECM involved in tumor progression and the immune-suppressive tumor microenvironment, allows tumor visualization. This represents novel opportunities for early diagnosis and potential monitoring of cancer progression. In addition, anti-Tenascin-C VHHs counteract the anti-adhesive properties of Tenascin-C on fibronectin substrate and restores the dendritic cell mobility inhibited by Tenascin-C in vitro [50].

Moreover, detecting and imaging ECM biomarkers has exciting potential for diagnosis. For example, NJB2, a VHH specific for an alternatively spliced domain of fibronectin, allows selective recognition of diseased tissues. Using noninvasive in vivo immune-PET/CT imaging, NJB2 can detect all stages of cancer progression, from neoplasia to metastasis, in multiple models of breast, melanoma, and pancreatic cancers. NJB2 also detects pulmonary fibrosis in a murine bleomycin-induced fibrosis model. Altogether, these results highlight ECM-targeted VHH as a promising candidate for VHH-based detection [51].

Similarly, the benefit of VHHs’ small size is their potential ability to cross the blood-brain barrier (BBB), which is one of the most important barriers for large molecules. The main challenge for novel therapeutics in the treatment of central nervous system (CNS) disorders is to engineer molecules that cross this barrier. VHHs emerge as promising therapeutic alternatives. The BBB is the major interface between the CNS and the peripheral circulation, formed by endothelial cells with tight junctions and adherent junctions. It tightly regulates the passage of molecules, ions, and cells between the blood and the CNS. The BBB’s restrictive nature is an obstacle to CNS drug delivery and excludes immunological molecules such as Abs. Fewer than 1% of all drugs are pharmacologically effective against brain disorders.

Conversely, VHHs targeting cerebral proteins have been found in mouse brains after systemic administration [52,53]. VHH trans-endothelial migration into the brain could be achieved through multiple routes. Some VHH naturally cross the BBB through adsorptive-mediated transcytosis that relies exclusively on the basic isoelectric point and cationic surface charges of the VHH. Only VHHs with a high isoelectric point will be able to cross the BBB by forming electrostatic interactions with the negatively charged cell membrane [53]. VHHs are also able to cross the BBB via the receptor-mediated transcytosis route. Multiple examples were investigated: VHH recognizing the transmembrane domain protein 30 A of α(2-3)-sialoglycoprotein expressed on endothelial cells [54]; VHH recognizing the insulin-like growth factor receptor (IGFR5) that recognizes IGF1R at the luminal side of endothelial cells. Their transcytosis across the cells and release in the brain environment on the abluminal side of the BBB were tested and reported [39].

As the VHH size is below the molecular weight limit for renal filtration (30 kDa), it is rapidly degraded by renal clearance, opening new avenues in diagnostics. However, this particularity may represent a disadvantage for long-term treatment. Indeed, the serum half-life of VHHs is only about 1 to 2 h, while mAb’s serum half-life is between 2 and 20 days, depending on IgG subclasses. Several strategies have been explored to modulate the pharmacokinetic profile of VHHs. First, some plasma proteins, such as serum albumin and IgG molecules, have naturally long half-lives. When conjugated to VHHs, they can slow down their clearance [43,55]. Indeed, Nanobody^®^ molecules comprising a VHH binding to albumin show a significant decrease in renal clearance compared to Nanobody^®^ molecule without. Such VHHs have a terminal half-life of 4.9 days in mice, corresponding to the expected albumin half-life (when administered intravenously in mice) [56]. Second, binding to albumin supports the targeting of inflamed tissues because the increased albumin metabolism in tumors and inflamed joints induces an albumin accumulation in these regions [44]. Finally, fusion of the VHH directly with the stabilizing group polyethylene glycol (PEG) molecules is one of the most commonly used methods. While VHH PEGylation slows down the blood clearance rate with high target site accumulation, it could also modify VHH affinity in some cases [45].

A particularity of VHHs is their long CDR3 loop, which allows good access for grooves and clefts on the surface of antigens [33]. Thus, VHHs represent great opportunities for untargetable, “hidden,” cryptic epitopes often inaccessible to conventional Abs. The major challenge to counteracting the coronavirus infectious disease 2019 (COVID-19) pandemic is the rapid mutation rate of the SARS-CoV-2 virus, limiting the efficacy of most vaccines and Abs developed [57]. Thus, it is essential to develop neutralizing Abs with broad-spectrum activity targeting multiple variants. In this context, VHHs represent an alternative strategy. Several VHHs recognizing conserved hidden epitopes of all SARS-CoV-2 Spike variants have been described, allowing the development of potent mutant-tolerant neutralizing VHHs to combat the coronavirus pandemic [58,59,60].

Another structural advantage of VHH is its hydrophilicity due to crucial amino acid mutations in the FR2 region, leading to better solubility compared to conventional VH [29]. While VHHs derived from conventional Abs were thought to be sticky and poorly soluble [61,62,63,64], substitution of a leucine for an arginine at position 45 confers greater solubility and stability to the VHH domain [3,33]. Some mutations occurring naturally during evolution allow the VHH domain to retain its full antigen-binding capacity.

VHHs’ small size and single polypeptide chain structure allow them to retain their conformation and their refolding capacity in extreme conditions, including high temperature, extreme pH, elevated pressure (500/750 MPa), proteolysis, and chemical denaturants (2–3 M guanidinium chloride, 6–8 M urea), making them suitable for different types of administration as drug [41,42,65,66,67]. While most VHHs exhibit good stability, phage display selection and additional modifications can enhance it. The incorporation of additional disulfide bonds between the CDR1 and the CDR3 of VHH increases their resistance to elevated temperatures and proteolytic degradation. Thus, VHHs demonstrate high thermostability up to 90 °C [4] and can maintain 80% activity at 37 °C for one week [33]. Substitutions in the VHH at the VH-VL interface are also responsible for stability at elevated temperatures by increasing the hydrophilicity of the fragment [40]. This suggests that VHH could be an ideal therapeutic agent that, when administered orally, resists the gastro-intestinal tract [68].

An important criterion sought for therapeutic fields is the ability to resist aerosolization and intranasal administration. VHHs are stable and robust enough to be administered as an aerosol directly into nasal epithelia and offer advantages for the treatment of pulmonary diseases [35]. Not only is the therapeutic agent delivered directly and in high concentration to the site of disease, but local administration should also ensure a rapid onset of therapeutic action. This is particularly important in acute disease settings, in which there is only a narrow window of opportunity for therapeutic intervention. For example, ALX-0171 is a trimeric Nanobody^®^ compound that binds and neutralizes Respiratory Syncytial Virus (RSV) [69]. Although it demonstrated antiviral efficacy even when nebulized into the airways/lungs of mice, providing a proof-of-concept for nebulized Nanobody^®^ molecule-based treatment [69], clinical trials in hospitalized children with RSV lower respiratory tract infection were stopped due to a lack of reaching therapeutic endpoints [18,70]. Recently, for COVID-19 disease, VHHs mVHH6 were found to neutralize SARS-CoV-2 even when aerosolized with heat shock and lyophilization [36].

Due to their unique intrinsic features, VHHs appear to have low immunogenicity. Indeed, VHHs are small, have rapid blood clearance, lack the Fc region (binding to Fc gamma receptors (FcγR) leading to immunogenicity), and share a significant degree of sequence similarity (95%) with human VH (VH3 gene family) [34,65,71]. While pre-clinical studies and clinical trials confirm the low immunogenicity of VHH-based treatments [72,73], a few clinical trials had to be terminated prematurely due to unwanted immune reactions and hepatotoxicity [74,75,76], suggesting that VHHs’ potential immunogenicity requires more understanding. Thus, we must remain cautious about the potentially low immunogenicity of VHHs, which seems to depend not only on the target protein but also on the patient’s history [31].

Together, VHH-specific features, combined with the lack of post-translational modification allowing them to be produced in large quantity, make VHHs exciting therapeutic opportunities. Translational research has developed numerous approaches leading to VHH as a next-generation therapy.

## 3. Nanobody^®^ Molecule, a Versatile Tool Emerging in Clinic

More than 150 mAbs have been successfully used in clinical practice. However, their large molecular weight and low tissue penetration may limit their efficacy. Due to their numerous advantages, VHHs are an attractive addition (Figure 3 and Figure 4), and to date, more than 30 Nanobody^®^ compounds are in clinical development [77,78,79] (Appendix A). Interestingly, the VHHs that are in clinical trials have different formats, including monovalent, bivalent, and multivalent structures (Figure 4A,B). Due to their small size and lack of a light chain, VHHs are ideal for fusion with other VHHs and/or with toxin molecules or other entities (Figure 4C). Moreover, although VHHs can penetrate deeply into tissues, their affinity for antigens and avidity can be increased by doubling the binding probability when fused with another VHH. VHH multimers can consist of VHHs targeting the same epitope (multivalent VHHs), thereby increasing avidity, VHHs targeting different epitopes of the same antigen (multiparatopic VHHs), thereby increasing specificity, or VHHs targeting different antigens (multispecific VHHs) to block independent signaling pathways simultaneously (Figure 4B). VHHs currently in clinical trials are well summarized in several reviews [77,79,80] as well as in Appendix A.

### 3.1. Nanobody^®^ Molecules in Imaging

Medical imaging is extensively used in the diagnosis of many diseases as an adjunct to clinical examination and tissue biopsies. In oncology, magnetic resonance imaging (MRI) and computer tomography (CT) scans are commonly used to assess tumor stages and look for metastasis and tumor response to treatment. However, these techniques only capture the anatomic features of the tumor without providing information at the cellular or molecular level. Noninvasive imaging approaches, such as positron emission tomography (PET), single emission computed tomography (SPECT), targeted contrast-enhanced MRI, and hybrid imaging techniques that combine multiple techniques, are widely used for diagnostic purposes. They use nonspecific radioisotopes based mostly on tumor cell glucose uptake and tumor cell proliferative ability and do not reflect tumor heterogeneity [81]. In terms of precision medicine, the ability to personalize treatment approaches is based on the cellular and molecular characterization of tumors. Therefore, targeted imaging has become essential for tumor stratification. The development of specific targeted molecules such as radiolabeled Abs and VHHs could address the challenge of target specificity. VHHs have all the requirements that an optimal noninvasive imaging agent should have, such as high specificity and sensitivity, as well as good tissue penetration, allowing rapid imaging after injection. Because of their short half-life in the bloodstream and rapid renal excretion, VHHs are suitable for rapid target labeling. They provide a better tumor-to-background signal ratio than conventional Abs, which accumulate in tissue, resulting in increased background and potential toxicity [81]. Several VHHs have been developed that target well-expressed transmembrane proteins, including EGFR, HER2, HER3, and CD38 [81]. For example, SPECT VHHs targeting HER2, which is expressed in approximately 20% of breast cancers, showed high tumor uptake, rapid blood clearance, low accumulation in non-target organs, and high tumor-to-blood and tumor-to-muscle ratios one hour after intravenous injection in HER2-positive mouse breast tumor models [82]. VHHs targeting HER2 in breast cancer have exciting potential, as observed in a phase I clinical trial. They target the organ of interest with little background in surrounding normal tissues and are rapidly cleared from the blood [83]. VHHs are also suitable for optical imaging, such as near-infrared (NIR) imaging using NIR fluorescent dyes. NIR imaging has been widely developed for molecularly guided precision surgery based on the imaging of specific markers [84]. While complete resection of cancer significantly improves favorable outcomes, distinguishing tumor tissue from benign surrounding tissue is still challenging. Thus, molecular-guided surgery could improve the rate of complete resection of cancer. Ab and VHH targeting cancer cell markers have shown enormous potential. For example, SGM-101, a 700 nm-conjugated mAb targeting carcinoembryonic antigen (CEA), a glycoprotein overexpressed in more than 90% of pancreatic cancers before and after neoadjuvant therapy, has been shown to allow visualization of pancreatic tumors during surgery [85,86,87]. However, one of the main disadvantages is the long interval between injection and surgery (3–5 days) due to the long circulation time of mAb in the blood. Thus, due to their short circulation time, VHHs are more suitable for non-invasive, short-term in vivo imaging than fluorochrome-labeled conventional mAbs and offer a promising technology for tumor-specific fluorescence delivery [88]. Studies have shown that ZW800-Fluorescent VHH targeting CEA facilitates visualization of cell lines and patient-derived pancreatic cancers, which better mimic a clinically relevant tumor microenvironment [84,89]. Similarly, a VHH conjugated with IRDye800 targeting carbonic anhydrase IX (CAIX), a specific membrane-bound protein that is upregulated under hypoxic conditions, was able to detect precancerous ductal carcinoma in situ (DCIS) [90]. While VHHs are promising tools for fluorescence-guided surgery, combining multiple probes and optimization strategies will improve imaging for tumor localization.

Since the advent of immunotherapy in cancer treatment, stratification of potential responders has been critical. Tumor expression of program cell death protein-1 (PD-1) and its ligand (PD-L1) can predict tumor response to immune checkpoint blockade. Tracing to non-invasively assess the expression of PD-1 and PD-L1 has been successfully performed using VHHs. In 2021, a clinical trial (Identifier: NCT05156515) using VHH APN09 labeled with the radionuclide PET (68Ga) targeting PD-L1 investigated the benefit of this alternative approach as a molecular imaging tracer for PET/CT scans. Intratumoral CD8 T cells within the tumor correlate with improved survival in response to immune checkpoint therapy. Radiolabeled (89Zr) VHHs specific for CD8 and immuno-positron emission tomography (PET) were able to distinguish anti-CTL4A-responsive from non-responsive tumors, suggesting that VHH and immuno-PET may represent a prognostic tool to predict patient response to checkpoint therapies [91]. The emergence of tumors resistant to immune checkpoint therapies has brought the next generation of immune checkpoints, such as Lymphocyte Activation Gene-3 (LAG-3), into the spotlight. Therefore, exploring their expression in tumors may provide guidance for therapy. Using radiolabeled VHH targeting LAG-3 and SPECT/CT scans, Lecocq. et al. studied the expression of LAG-3 on tumor-infiltrating lymphocytes (TILs) in several syngeneic mouse cancer models and showed that tumors harboring LAG-3-expressing TILs benefit from a combination therapy of PD-1/LAG-3 blocking Abs [92,93].

Among the many potential applications of VHHs, their use in medical imaging is one of the most promising. Because of their affinity and short half-life, they could improve imaging and patient comfort while reducing the risk of toxicity. They could also provide a new tool for non-invasive monitoring of protein expression dynamics to predict therapeutic outcomes.

### 3.2. Nanobody^®^ Molecules, from Bench to Bedside (or Path to the Way of Alternative Therapeutics)

The in-depth study of Nanobody^®^ molecules and their multiple advantages has led to the development of potential therapeutic alternatives for many diseases.

#### 3.2.1. Nanobody^®^ Molecules in Central Nervous System Diseases

Although not all VHHs can passively cross the BBB, their ease of engineering and the creativity of researchers have allowed the development of several alternative strategies for neuronal malignancies. Alzheimer’s disease (AD) is the most common form of dementia in adults. It is estimated to affect more than 30 million people worldwide, with the number of people affected expected to double in the next 20 years. It is characterized by two major brain changes: the accumulation of amyloid-β (Aβ) peptides leading to extracellular senile plaques and the accumulation of intracellular hyperphosphorylated tau proteins called neurofibrillary tangles (NFTs) [94]. Drug development targeting Aβ has failed in the clinic, and there are currently no treatments that delay or halt disease progression [94]. The use of VHHs to treat AD has been investigated and several VHHs have shown therapeutic potential in vitro [95]. VHH R3VQ, which targets Aβ deposits, was shown to extravasate from the blood into the brain parenchyma and to visualize specific amyloid deposits in a mouse model with β-amyloid lesions [96]. Although Aβ is the main target for targeted AD therapy, tau proteins have been considered alternative targets. VHH A2, which targets tau inclusions, was able to label NFT-like structures in neurons from Tg4510 mice (a genetic model with NFT) after intravenous administration [96]. In addition, VHH Z70, which recognizes the PHF6 sequence known for its ability to nucleate, effectively inhibited tau aggregation in vitro. Based on these results, the ability of VHH Z70 to block tau seeding was examined in a THY-Tau30 transgenic mouse model by injection into the hippocampus. It was found that the delivery of VHH needs to be optimized, as only direct intracellular expression of VHH Z70 by lentiviral vectors reduced tau seeding pathology [97]. Thanks to the ease of VHH engineering, one way to optimize VHH BBB penetration is a gene transfer strategy based on AAV vectors that overcome the BBB to deliver VHH directly to the brain [98]. As a proof-of-concept, the potential of AAV-administered VHH to inhibit BACE1, a well-characterized target in AD, was investigated. This showed that a single systemic dose of AAV-VHH-B9 targeting BACE1 had positive long-term effects on amyloid load, neuroinflammation, synaptic function, and cognitive performance in the AppNL-G-F Alzheimer mouse model [99]. Another CNS-related disease in which VHH has been studied is Parkinson’s disease (PD). This pathology is most associated with intracellular inclusions called Lewy bodies [100], which contain misfolded, aggregated α-Synuclein (α-Syn), a neuronal protein that regulates neurotransmitter release [100]. The bifunctional VHH targeting the C-terminus of α-Syn fused to a proteasome targeting signal, NbSyn87PEST, can degrade α-Syn in neuronal cells in vitro [101]. Similarly, VHH targeting the N-terminal region, Nbα-syn01, has been identified by Phage display screening of the library against monomeric α-Syn. The affinity of the monovalent Nbα-syn01 and the constructed bivalent format, BivNbα-syn01, showed that Nbα-syn01 and its bivalent format were able to inhibit α-Syn aggregation in vitro. Another approach aimed to specifically target pathogenic preformed α-Syn fibrils. To do this, extracellular disulfide bond-free synthetic VHH libraries in yeast have been performed, which enabled the identification of PFFNB2 VHH that recognizes α-Syn-preformed fibrils but not α-Syn monomers [100]. To counteract the low BBB crossing ability of PFFNB2 VHH, Adeno-associated virus (AAV)-encoding EGFP fused to PFFNB2 (AAV-EGFP -PFFNB2) has been engineered and was able to disrupt and destabilize the structure of existing α-syn preformed fibrils, thus preventing the spread of α-syn pathology to the cortex in transgenic mice mimicking this disease [100]. Another distinctive feature of VHH is its preference for clefts and furrows, which allows the development of new pharmaceutical alternatives for “hard-to-reach proteins” such as GTPase. In PD patients, the PD-associated protein Leucine-Rich Repeat Kinase 2 (LRRK2) is a key player that is often mutated or overactivated, leading to a decrease in GTPase activity and an increase in kinase activity. Drug development strategies have focused on the development of ATP-competitive LRRK2 kinase inhibitors. VHHs that bind the bacterial LRRK2 homolog in a conformation-specific manner significantly increase GTP turnover through allosteric modulation. This represents a potential new strategy to overcome the effects of LRRK2 PD mutations [102].

#### 3.2.2. Nanobody^®^ Molecules in Immune System Disorders

Immune system disorders are a broad spectrum of related diseases characterized by immune system dysregulation leading to activation of immune cells to attack autoantigens, resulting in inappropriate inflammation and damage to multiple tissues. The pleiotropic cytokine tumor necrosis factor alpha (TNF) is implicated in several inflammatory diseases, as inappropriate or excessive activation of TNF-α signaling is associated with chronic inflammation and may eventually lead to the development of pathological complications [103]. Several anti-TNF inhibitors, including mAbs, have been developed and are currently used for a wide range of autoimmune diseases [103]. However, mAb-based therapies are available as intravenous (IV) or subcutaneous (SC) formulations because their high molecular weight and degradation in the stomach preclude oral administration [104]. In addition, there are various side effects such as headaches, skin rashes, anemia, infectious diseases, and most commonly, a reaction at the injection site through a SC route [103]. Therefore, VHH-based therapy has enormous potential for inhibiting inflammation in patients. Biparatopic anti-TNFR1 VHHs conjugated with anti-albumin VHHs inhibit TNF expression and reduce inflammation in ex vivo cultures of colon biopsies from patients with Crown disease [105]. In addition, VHHs targeting TNF may be suitable for the treatment of rheumatoid arthritis. Bispecific VHHs against TNF were 500-fold more effective in inhibiting inflammation than the corresponding monovalent VHHs [106]. Moreover, this inhibitory potential exceeds that of commercial anti-TNF mAbs used clinically. A phase III clinical trial in which ozoralizumab, a humanized trivalent bispecific Nanobody^®^ molecule (now approved by Taisho as Nanozora^®^) consisting of two anti-TNF VHHs and one anti-human serum albumin VHH, was combined with methotrexate (MTX) in MTX-resistant rheumatoid arthritis patients showed a significant decrease in disease symptoms compared to patients receiving MTX alone (JapicCTI identifier: 184029). Since TNF is a major proinflammatory cytokine in inflammatory bowel disease (IBD), a chronic immune-inflammatory disease of the gastrointestinal tract, anti-TNF therapy is effective in a proportion of patients. VHH V565 was isolated from a phage library prepared from lymphocytes of a human TNF-immunized llama and engineered to be resistant to intestinal and inflammatory proteases. After oral administration, V565 was present in the intestinal lumen and was able to bind and neutralize TNFα and reduce the inflammatory process in four patients with terminal ileostomy [107].

#### 3.2.3. Nanobody^®^ Molecules in Infectious and Viral Diseases

Parasitic infections in humans are a worldwide public health problem. For example, *Trypanosoma brucei*, which causes African sleeping sickness in humans, is caused by the parasitic flagellate *Trypanosoma brucei*, which is injected into the body by the tsetse fly. The dense coat of variant surface glycoproteins (VSG) of African trypanosomes provides the primary interface between host and pathogen. High-affinity VHHs against conserved cryptic epitopes of VSG in conjugation with a truncated version of the human trypanolytic protein APOL1, which prevents neutralization by trypanosomes, are potent trypanolytic agents [108]. In addition, high doses of anti-VSG VHH can bind deep inside the envelope of trypanosomes and inhibit endocytosis, resulting in potent anti-parasite effects in vivo [109]. Similarly, malaria remains one of the most prevalent parasitic diseases in the world, caused by six species of *Plasmodium* parasites, of which *Plasmodium falciparum* is the most lethal. Invasion of host erythrocytes occurs in several steps, including initial attachment mediated by the merozoite surface protein, formation of tight junctions mediated by erythrocyte-binding proteins and reticulocyte-binding proteins, and invasion. One strategy to combat malaria is to block erythrocyte infection. Epitopes that neutralize invasion have been identified via the development of invasion-inhibiting VHHs that target the PCRCR complex, essential for merozoite invasion of erythrocytes [110]. Another strategy that has been exploited is to prevent transmission of malaria parasites from mosquitoes to humans by inhibiting infection of the parasites in the mosquito [111]. Indeed, fertilization of the malaria parasite occurs in the midgut of the female Anopheles mosquito, so inhibiting fertilization stops transmission of the parasite from mosquito to human. Surface-associated proteins play a critical role in the life cycle of the Plasmodium parasite. In particular, the 6-cysteine protein family is expressed during the sexual phase. Pfs230 is the largest member of the 6-cysteine protein family and is expressed on the surface of gametocytes and gametes. The use of non-overlapping Pfs230 VHHs significantly blocked *P. falciparum* transmission and reduced the formation of exflagellation centers [111]. To survive in Anopheles mosquitoes, Plasmodium parasites must evade the mosquito’s immune response. PIMMS43 (*Plasmodium* Infection of the Mosquito Midgut Screen 43), an ookinete and sporozoite surface protein, is required for parasite evasion of the Anopheles immune response. Disruption of PIMMS43 in *P. berghei* leads to ookinete elimination in the mosquito midgut. mAbs that bind PIMMS43 significantly reduce the prevalence and the intensity of infection [112]. Thus, a strategy to eradicate malaria disease is to generate genetically modified mosquitos expressing PIMMS43-blocking VHH. Indeed, if released in nature, these mosquitoes could rapidly spread their nanobody-expressing genes to wild mosquitoes, rendering them resistant to parasite infection.

Among infectious diseases, acquired immunodeficiency syndrome (AIDS) caused by human immunodeficiency virus (HIV) infection, is still one of the deadliest infectious diseases in the world. Bivalent and trivalent VHH constructs (gp120 and gp41) targeting different epitopes on the envelope glycoproteins Env, the protein that forms the viral envelope, showed higher neutralizing efficacy than single VHHs against viruses of specific clades [90]. As VHH are simple, rapid, and inexpensive to produce, they allow fast adaptation for rapidly evolving pathogens such as influenza and coronaviruses. Thus, virus-neutralizing VHH has been developed to several animal and human virus families [18]. Since the COVID-19 pandemic, biparatopic and multivalent VHHs have been developed to tackle SARS-CoV-2 infections. Recent advances in the coronavirus structure have shown that the surface-exposed portion of the Spike is composed of two subunits: the S1 domain that binds to ACE2 through the receptor binding domain (RBD) and the S2 domain that drives fusion of the viral and host cell membranes. Numbers of VHHs have been identified to target the RBD domain of the Spike protein on the surface of SARS-CoV-2 [113]. Bivalent, bispecific, and trivalent VHHs against the SARS-CoV-2 receptor binding domain are efficient in competing with ACE-2 and demonstrating good therapeutic effects in ex or in vivo experiments [36,58,114,115,116,117]. Furthermore, studies have shown that VHHs neutralize wild-type SARS-CoV-2 and different variants, including delta D614G variants in the nanomolar range, demonstrating their potential as antiviral agents [57,118]. In addition, when fused with another VHH that binds to a distinct epitope on the same domain, they enhance the neutralization of the protein [118]. Fusion with the Fc domain of human IgG1 also increased efficacy [58,114,118,119]. Moreover, a 4-armed polyethylene glycol (PEG) tetrameric VHH construct, designed to neutralize SARS-CoV-2, enhances neutralization compared to the dimeric construct and the VHH-Fc fusion variant [120]. The pandemic highlighted VHH as an ideal tool for defeating infectious disease. Indeed, their fast and strong production efficiency allows quick reactivity, which is critical to counteracting highly evolving diseases. In addition, their aerosolization potential makes them particularly suitable for air-transmissible viruses and thus essential for prophylactic alternatives. Finally, these simplified single-gene-encoded VHHs are quite easy to manufacture and can be produced in plants and in yeast for bulk applications, enabling global coverage. Thus, VHH development may be an ideal strategy to address future health crises [121].

Infectious and parasitic diseases can also be transmitted through foodborne pathogens that cause significant morbidity and mortality. Due to their high sensitivity, immunoassays, especially those based on VHHs, are being used in an increasing number of studies to detect toxins and pathogens [122]. For example, *Salmonella enteritidis* (*S. enteritidis*), being one of the most prevalent foodborne pathogens worldwide, poses a serious threat to public safety [123]. VHH-based *sandwich ELISA* assay to detect *S. enteritidis* in the spiked milk and monitor the distribution in the challenged chicken has proven efficacy [123]. More generally, the identification and treatment of plant pathogens could be useful for agronomic applications and take agriculture into a new dimension. For example, Zucchini rottermosaic virus (ZYMV), an aphid-borne potyvirus that affects all cucurbits and causes significant agronomic problems, can be identified with high specificity by using VHHs directed against the coat protein [124]. Tomato leaf curl Sudan virus (ToLCSDV), the most common begomovirus infecting tomatoes in Saudi Arabia, can also be detected using VHH-based technology [124]. In addition to detection methods, VHH can also be developed as a therapeutic agent. For example, degenerative fanleaf disease, caused by the icosahedral grapevine fanleaf virus transmitted by nematodes, is a major problem for the grape industry. Specific VHHs for GFLV have been discovered and shown to neutralize the virus [125].

#### 3.2.4. Nanobody^®^ Molecules in Oncology

Conventional Ab-based targeted cancer therapy has proven efficacy, with the potential to reduce toxic side effects to healthy cells surrounding tumor cells compared to chemotherapy. However, the full potential of Abs is limited due to their large size, low stability, slow clearance, and high immunogenicity [126]. In cancer research, VHHs were first used to target transmembrane proteins that are overexpressed on tumor cells. For example, Epidermal growth factor receptor (EGFR, also known as HER1) is already targeted by mAbs (cetuximab) in combination with chemotherapeutic measures. In 2007, an antagonistic anti-EGFR VHH was developed to efficiently inhibit EGF binding to EGFR and its downstream signaling. This led to the inhibition of tumor cell proliferation in vitro and in vivo [127]. Later, a biparatopic VHH was designed and showed better EGFR blocking capacity than the monovalent construct. In vivo, the biparatopic VHH exerts inhibition as efficient as mAb cetuximab in a model of athymic mice bearing A431 cell line xenografts [128]. More recently, the improved efficiency of biconstructive VHHs has also been demonstrated by conjugating two anti-EGFR VHHs targeting distinct non-overlapping epitopes and an anti-mitotic agent, the monomethyl auristin E. This tetravalent VHH drug showed a high conjugation efficiency, a small binding interface, and highly potent antitumor activity [129].

A hallmark of cancer is the alteration of the ECM surrounding the tumor, which differs significantly in architecture and composition from that of normal tissue. Tumor ECM is more abundant, denser, and stiffer than healthy tissue, which profoundly alters its response to therapy [130]. Drug distribution in the tumor occurs by diffusion [37], as ECM acts as a physical barrier that impedes drug transport. While efficacy remains limited, ECM softening improves response to chemotherapy and immunotherapy. VHH could represent a novel and complementary strategy to tumor softening.

For overcoming resistance to inhibitors, multivalent VHH is of great interest. A multivalent structure comprised of an anti-EGFR VHH and two anti-HER2 affibodies coupled with Adriamycin, a chemotherapeutic drug, showed high anti-tumor activity both in vitro and in vivo [131]. Affibody^®^ molecules are single protein chains derived from staphylococcal protein A imitating monoclonal Abs. To overcome immunotherapy tumor resistance and relapse, a bispecific anti-PD-L1/VEGF VHH was also designed. It exhibited efficient inhibition of cell proliferation and progression of angiogenesis in vitro and ex ovo, which is encouraging for future cancer therapies [132].

Bispecific T-cell engagers (BiTEs) are a new class of immunotherapeutic molecules intended for the treatment of cancer. The generation of BiTes targeting Vγ9Vδ2-T-cells, a small subset of the T-cell population, and EGFR showed T-cell activation and subsequent tumor cell lysis via secretion of inflammatory cytokines both in vitro and in vivo [133]. Interestingly, tumor cell lysis was independent of *KRAS* and *BRAF* mutation status. This contrasts with anti-EGFR monoclonal Ab therapy, which is associated with intrinsic and acquired resistance. In the case of B-cell leukemia and lymphomas, relapse is common due to tumor cell changes during targeted therapies. A bispecific VHH that targets CD20 and HER2 can kill tumor cells that express these antigens at the surface. It could be an alternative to conventional targeted therapy to prevent antigen escape [134].

One way to decrease chemotherapy off-target cytotoxicity is to deliver the chemo-agent directly into the tumor. Ab-based drug conjugates (ADC) have shown promising treatment efficacy compared to conventional chemotherapy [135], and several are in advanced stages of clinical testing [136] or already approved by the United States Food and Drug Administration US-FDA [137]. ADC comprises a mAb, which can be covalently conjugated to chemodrugs and toxins through chemical linkers and will act as a carrier to bring a drug to the targeted tumor cells. This strategy will help reduce systemic chemodrug concentration. Consequently, cytotoxic side effects will be limited in patients undergoing sustained treatment, as the drug will go in the right place and will target specific delivery. This is a tremendous advantage for cancer therapy, as chemotherapy is often associated with strong and difficult-to-handle side effects, often resulting in treatment termination. However, the large size of mAb, combined with the abnormal tumor vessels, limit ADC tumor penetration and thus efficacy [138]. The small size, monomeric structure, and robustness of VHHs make them ideal drug carrier alternatives [139]. VHH-based drug conjugates (NDC), conjugating VHH, and chemotherapeutics have been investigated, allowing targeting of tumor cells with high affinity and low side effects. For example, CD147 is a glycosylated transmembrane protein overexpressed in pathological tissues and involved in cancer progression and drug resistance. The anti-CD147 VHH conjugated with doxorubicin displayed good antitumor efficacy for the 4T1 breast tumor model in Balb/c mice [140]. Similar strategies have been employed with an anti-EGFR and an anti-Her2 VHH conjugated with doxorubicin and showed good anti-tumor activity in vivo [141]. VHHs targeting CDH17, a protein up-regulated in human gastric cancer, showed not only efficient recognition but also cytotoxic activity and tumor-inhibitory activity in vitro and in vivo when fused with the modified *Pseudomonas endotoxin A* lacking N-terminal cell-binding domain (PE38) [142]. Fusion of a VHH directed against VEGFR2 and the *Pseudomonas Endotoxin A* led to high specificity and effective tumor inhibition [143]. Similarly, a combination of VHHs with *Enterobacter cloacae beta-lactamase* activates a co-administered anticancer prodrug, the cephalosporin nitrogen mustard prodrug 2-(4-carboxybutanamido)-cephalosporin, to target cancer cells with carcinoembryonic antigen [71].

Thus, due to their small size, high affinity, and ease of engineering, VHH-based therapies will emerge, allowing tailored suited treatments and personalized medicine for a wide range of patients and pathologies.

## 4. Perspectives

Although VHHs were discovered as early as the 1990s, the enthusiasm they have generated from a scientific and clinical point of view is still quite recent. In fact, their discovery initially went unnoticed; it was not until 2012 that the scientific world became excited about VHHs. Ablynx, a spin-off of the Vlaams Institute Voor Biotechnologie (VIB) and the Free University of Brussels (VUB), quickly (2018) became Sanofi’s Nanobody platform. Ablynx is involved in the discovery and development of nanobodies, specifically Cablivi^®^ (caplacizumab-yhdp).

Despite major interest from biopharmaceutical companies, VHHs’ limitations could be anticipated, potentially restraining their applications. First, their mechanism of action depends on their pharmacokinetic properties. They present a short half-life in vivo through rapid excretion via the kidneys. This half-life had to be extended to improve target engagement. Nowadays, half-life extension strategies, such as albumin module, increase this VHH half-life but also modulate their valence thus potentially decreasing their efficacy. Thus, even though we can now use VHHs in the clinic, limitations could be envisioned regarding pharmacokinetics. Second, VHHs remain lama/camelids Abs used in humans. As such, major efforts are needed to humanize VHHs to avoid the adverse effect of anti-drug Abs development. The required modifications are time-consuming, expensive, and may limit VHH applications. Third, once large-scale production is required for VHHs’ therapeutic use, one may envision further drawbacks. VHHs’ specific individual properties, such as critical micelle concentration (CMC), may indeed impede manufacturing. CMC evaluation is required to avoid the formation of micellar aggregates while VHHs are in production. Finally, VHH side effects will depend on the chosen target. Nonetheless, because of their benefits and versatility, the therapeutic and industrial applications of VHHs are plentiful. VHHs will not only be the focus of many future therapies but also of many industrial developments. VHHs’ low-production cost combined with remarkable stability and ease of use are of great interest for industries to develop applications in various fields.

From a more philanthropic perspective, VHHs have several advantages that could benefit most countries worldwide. The cross-neutralizing of HCAbs against SARS-CoV-2 was found in camels that were seropositive but free of Middle East respiratory syndrome coronavirus (MERS-CoV) [144]. These hyperimmune camels may represent an important source of therapeutic agents for the prevention and treatment of COVID-19. Because HCAbs are present not only in blood but also in all biological fluids of camels, including milk and urine, prophylactic and therapeutic agents can be produced and distributed in isolated populations. Since VHHs are also very stable in harsh environments, another interesting possibility is to produce VHH in plants, including roots, seeds, leaves, and fruits. This type of large-scale production has two advantages: First, it is an easily accessible way to produce VHHs. Second, while most of the conventional Abs are still produced in animals, this type of production is animal-free. To conclude, VHHs are powerful tools but still under development. They represent the next generation of therapeutic solutions with incredible potential applications. This improbable discovery, which will tremendously change diagnostic, therapeutic, and industrial perspectives in the future, highlights the role of experimentation and basic research.

## Figures and Tables

**Figure 1 ijms-24-13229-f001:**
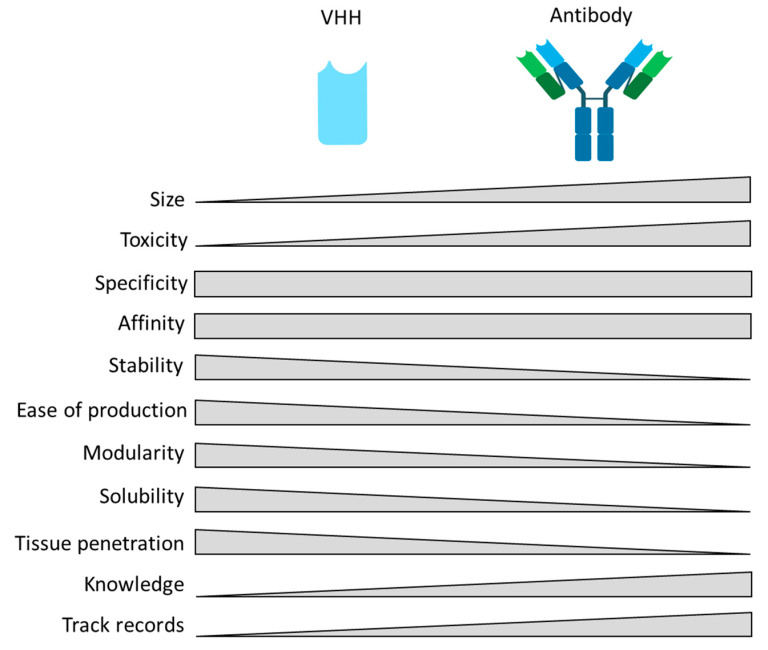
Comparison of VHH and antibodies (created with BioRender.com accessed on 20 February 2023).

**Figure 2 ijms-24-13229-f002:**
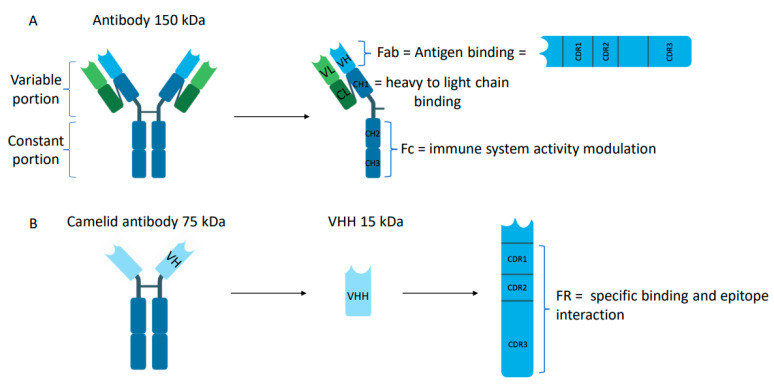
Structural difference between antibody and VHH. (**A**). Conventional antibody structure and heavy chain structure (**B**). Camel HCAb structure and VHH structure (created with BioRender.com, accessed on 20 February 2023). VH: heavy chain; VL: light chain; CL CHs: constant domains; CDRs: complementary determining regions; Fc: fragment crystallizable; Fab: fragment antigen-binding; VHH: camelid heavy-chain variable domains; FRs: frameworks.

**Figure 3 ijms-24-13229-f003:**
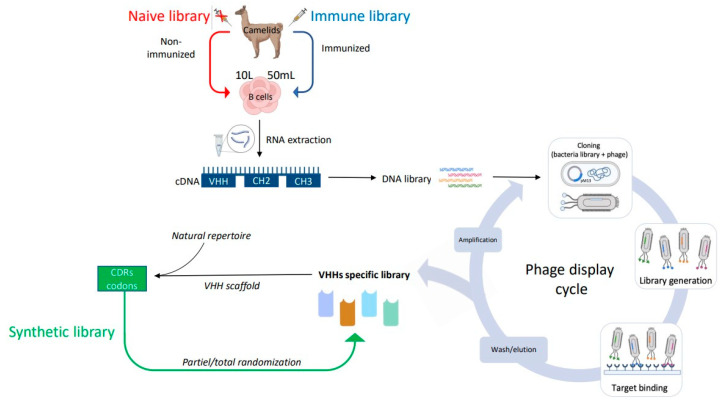
Schematic overview to generate immune, naive, and synthetic VHHs libraries through phage display (created with BioRender.com, accessed on 20 February 2023). RNA: Ribonucleic acid; cDNA: complementary DNA; VHH: camelid heavy-chain variable domains; CHs: constant domains; CDRs: complementary determining regions.

**Figure 4 ijms-24-13229-f004:**
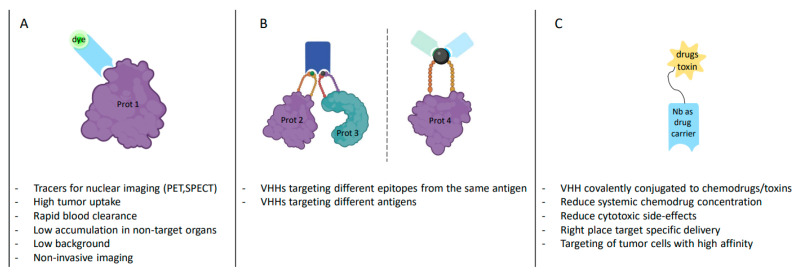
VHH, a versatile tool (**A**). Monovalent VHH (**B**). Multivalent VHH: bispecific/biparatopic (**C**). VHH drug carrier (created with BioRender.com, accessed on 20 February 2023). Prot: protein; VHH: camelid heavy-chain variable domains; PET: positron emission tomography; SPECT: single-photon emission computed tomography.

**Table 1 ijms-24-13229-t001:** Summary table of main VHH properties.

Properties	VHH	Conventional Antibody	References
Immunogenicity	Not yet well documented, needs to be more investigated.	Often	Ackaert, C., et al., 2021 [31]
Thermostability	Up to 90 °C	<90 °C	Van der Linder, R., et al. 2000 [32]; Arbabi Ghahroudi, M., et al. 1997 [33]; Kunz, P., et al., 2018 [34]
Nasal administration	Aerosol	No	Rohm, M et al., 2017 [35]; Schoof, M., et al., 2020 [36]
Size	15 kDa	150 kDa	Hamers-Casterman, C., et al., 1993 [3]
Tissue penetration	BBB *, ECM **, grooves, clefts	No	Subrahmanyam, N., et al., 2021 [37]; Debie, P., et al. 2020 [38]; Pothin, E., et al., 2020 [39]
Stability/solubility	High stability and solubility	Soluble and stable with risk of aggregation	Davies, J., et al., 1996 [40]; Dumoulin, M., et al., 2009 [41]; Harmsen, M., et al., 2007 [42]
Half life	Fast clearance but could be modulated by e.g., adding an albumin-binding VHH, an Fc domain, or Polyethylene glycol ((PEG)-ylation)	Long (Fc-mediated)	Konterman, R.E., et al., 2011 [43]; Wunder, A., et al., 2003 [44]; Fishburn, C.S., et al., 2008 [45]

* BBB: Blood-Brain Barrier; ** ECM: Extra-Cellular Matrix.

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
