# Peer review of "NANOBODY^®^ Molecule, a Giga Medical Tool in Nanodimensions"

_ijms, 2023, doi:10.3390/ijms241713229_

Round 1
Reviewer 1 Report
The authors present an enthusiastic perspective on the clinical potential of nanobodies with examples from in vitro and in vivo studies. Although there have been similar review articles published, the field appears to be rapidly expanding and hence this review can be considered as timely. The extensive bibliography for following up on details of the work cited is helpful. A major theme of the article is a comparison between nanobodies and whole antibodies and how nanobodies address what are described as limitations of antibodies for clinical use. However, antibodies are very successful drugs, and the biopharmaceutical industry has a long track record and knowledge base for the development of antibodies. This is not yet the case for nanobodies. Nanobodies should be viewed as an alternative, bringing different advantages but not a substitute. Therefore, a more balanced narrative is required in this article. Although some of the limitations of nanobodies are discussed in the final section, a more critical assessment of the literature that is cited would make this review more valuable. There are currently only two clinically approved nanobody based drugs.
Specific comments
1. Adding a table summarising the development stage of nanobody products that are in advanced pre-clinical/clinical testing would be very useful.
2. Some aspects of clinical applications (e.g. in oncology) appear in the sections on technology and advantages of nanobodies. It is suggested that these earlier sections are reviewed and opportunities to rationalise these sections in relation to the later disease-focused ones is considered.
3. The title lacks real meaning making use of “buzz” words, and therefore a more informative title should be considered.
Minor grammatical errors only suggest read through by native English speaker.
Reviewer 2 Report
I thank the authors for an interesting review of Nanobody® molecules, which are more promising than conventional antibodies. Nanobody® represents the next generation of therapeutic solutions with incredible potential applications.
There are major notes that should be fixed:
1. The authors do not pay enough attention to the comparison of Nanobody® with other types of therapeutic antibodies. For example, is there an advantage of Nanoantibodies over scFv?
2. In the introduction, we should dwell in more detail on the differences in the antibody structure of camels and humans. Describe the beneficial properties of Nanobody® in more detail
A few minor notes:
Refer to Figures 2A, 2B, 4A, 4B, 4C
Line 76: What is the structure of the other 50% camel antibodies? It should be mentioned.
Line 78: Please provide more details on the isolation of camel antibodies by affinity chromatography.
Line 101: All types of therapeutic monoclonal antibodies should be mentioned. About the advantages and disadvantages of each.
Line 150: not only VHH and Conventional antibody should be compared, but also other types of therapeutic antibodies (scFv, Fab and others)
Lines 158-170. The paragraph has no context. This data should be merged with section 3.2
Round 2
Reviewer 1 Report
The authors have addressed the main concern about nanobodies vs antibodies in their revised submission and provided a very useful table on the pre-/clinical development of nanobody based therapeutics. The proposed revised title works well.